# Anti-senescent drug screening by deep learning-based morphology senescence scoring

Dai Kusumoto[1,2], Tomohisa Seki[3], Hiromune Sawada[1], Akira Kunitomi[4], Toshiomi Katsuki[1], Mai Kimura[1], Shogo Ito[1], Jin Komuro[1], Hisayuki Hashimoto[1,2], Keiichi Fukuda [1] & Shinsuke Yuasa [1✉]

Advances in deep learning technology have enabled complex task solutions. The accuracy of image classification tasks has improved owing to the establishment of convolutional neural networks (CNN). Cellular senescence is a hallmark of ageing and is important for the pathogenesis of ageing-related diseases. Furthermore, it is a potential therapeutic target. Specific molecular markers are used to identify senescent cells. Moreover senescent cells show unique morphology, which can be identified. We develop a successful morphology-based CNN system to identify senescent cells and a quantitative scoring system to evaluate the state of endothelial cells by senescence probability output from pre-trained CNN optimised for the classification of cellular senescence, Deep Learning-Based Senescence Scoring System by Morphology (Deep-SeSMo). Deep-SeSMo correctly evaluates the effects of well-known anti-senescent reagents. We screen for drugs that control cellular senescence using a kinase inhibitor library by Deep-SeSMo-based drug screening and identify four anti-senescent drugs. RNA sequence analysis reveals that these compounds commonly suppress senescent phenotypes through inhibition of the inflammatory response pathway. Thus, morphology-based CNN system can be a powerful tool for anti-senescent drug screening.

---

[1] Department of Cardiology, Keio University School of Medicine, 35 Shinanomachi, Shinjuku-ku, Tokyo 160-8582, Japan. [2] Center for Preventive Medicine, Keio University School of Medicine, 35 Shinanomachi, Shinjuku-ku, Tokyo 160-8582, Japan. [3] Department of Healthcare Information Management, The University of Tokyo Hospital, 7-3-1 Hongo, Bunkyo-ku, Tokyo 113-8655, Japan. [4] Center for iPS Cell Research and Application, Kyoto University, Kyoto 606-8507, Japan. ✉email: yuasa@keio.jp

Advances in deep learning technology have enabled complex task solutions[1]. The accuracy of image classification has increased rapidly owing to the development of convolutional neural networks (CNNs)[2,3]. CNNs have been applied to broad medical research fields[4], and image classification is employed as a diagnostic tool in the clinic[5]. In the biological field, cell morphology images obtained by phase-contrast microscopy contain numerous biological data such as cellular identity and status, which are currently evaluated by molecular biology techniques. A morphology-based identification system using CNN can replace the molecular biology techniques in some tasks and be applicable to various research areas. We previously developed a label-free system to identify endothelial cells among various cell types derived from induced pluripotent stem cells by phase-contrast microscopy images using a CNN[6]. Many reports demonstrate the high potential of CNNs in a classification or identification task. Versatile biologic systems should construct quantitative and not just qualitative classifications[7]. CNNs are a potential tool to develop non-biased quantitative evaluation systems.

Endothelial cells serve many functions in homoeostasis and diseases. Cellular senescence plays an important role in age-related diseases. Endothelial cells are pivotally involved in the pathology of age-related diseases through cellular senescence. Endogenous and exogenous stresses such as reactive oxygen species (ROS), telomere dysfunction, DNA damage, inflammatory cytokines, and drugs such as anti-cancer drugs, induce cellular senescence[8]. Senescent cells show an inflammatory phenotype called senescence-associated secretory phenotype (SASP) and contribute to age-related disease progression[9]. Cellular senescence is considered a potential therapeutic target for age-related diseases[10,11]. Thus, drugs that directly intervene in endothelial cell senescence may represent a therapeutic option. Specific biological markers are commonly used for cellular senescence screening such as senescence-associated beta galactosidase (SA-β-gal), P16, and P21. Cellular senescence can also be defined by specific morphology such as flat and enlarged cell bodies and heterochromatin aggregation[12]. Despite this, the unbiased quantitative evaluation of those morphological changes for a large number of cells is difficult in using conventional methods. A scoring system that can quantitatively assess the cellular state could be an important tool for drug screening.

In this study, we developed a robust, morphology-based CNN system to identify senescent cells. Additionally, we established an automated, non-bias quantitative scoring system to evaluate the state of endothelial cells using senescence probability output directly from pre-trained CNN, Deep Learning-Based Senescence Scoring System by Morphology (Deep-SeSMo) (Supplementary Fig. 1a). Deep-SeSMo-based drug screening using a kinase inhibitor library was used to identify anti-senescent drugs.

## Results

### High accuracy identification of senescent cells by a CNN.
We induced cellular senescence in human umbilical vein endothelial cells (HUVECs) by using three different stressors: ROS, an anti-cancer reagent, and replication stress (Supplementary Fig. 1b). Hydrogen peroxide ($H_2O_2$), camptothecin (CPT), and repetitive passage (replication: rep)-induced cellular senescence was confirmed by SA-β-gal activity (Supplementary Fig. 1c, d). In senescent cells, the P21–P53 pathway is activated to induce cell cycle arrest[13]. Expression of *P21*, a marker of cell senescence, was also upregulated in senescent cells (Supplementary Fig. 1e). Next, we prepared $50 \times 50$ pixels of input datasets at the single-cell resolution level from phase-contrast images (Supplementary Fig. 2a, b). Senescence was independently induced four times for

each stress type to increase the data generalisability. For each induction, 10 phase-contrast images were acquired under each condition, and the number of obtained images was 92,242 for $H_2O_2$-induced senescence, 41,207 for $H_2O_2$ control, 134,097 for CPT-induced senescence, and 64,535 for CPT control (Fig. 1a and Supplementary Fig. 2c, d). The images were then analysed in a network to predict them as senescence or control (Supplementary Fig. 2e). The predictions were compared with predetermined answers, and weights were automatically and iteratively optimised to train the CNN and thereby increase accuracy. We examined whether the CNN could classify the senescent cells induced by either $H_2O_2$ or CPT, and control cells. After training, the CNN could classify $H_2O_2$- or CPT-induced senescent cells and control cells with high accuracy (Supplementary Fig. 3a–d). Next, we mixed $H_2O_2$- and CPT-induced senescence images and trained the CNN to classify senescent and control cells. The CNN was successfully trained and showed no discrepancy between the loss values in training and validation data (Fig. 1b and Supplementary Fig. 3e). The trained CNN performed strongly; the accuracy, *F*1 score, and area under the curve (AUC) of the receiver operating characteristic (ROC) were 0.93, 0.88, and 0.98, respectively (Fig. 1c, d). We compared these results with feature-based traditional machine learning methods (Support Vector Machine, Random forests, and logistic regression) to examine the superiority of CNN. To analyse cellular images using classical machine learning models, we extracted features of images to create input datasets. We used Histograms of Oriented Gradients (HOGs), which is one of the most commonly used feature descriptors and trained the machine learning models. The accuracy and *F*1 score of traditional machine learning models were lower than that of the CNN, and we concluded that the CNN is the most suitable method for our study (Supplementary Fig. 3f).

### CNN generalisability.
Generalisability is important in machine learning and requires external validation of the analysis. To confirm whether CNN could identify senescent cells in the datasets of different senescence induction methods, we acquired new datasets not used for the CNN training. We prepared datasets for the three induction methods, and each induction was independently performed three times. These images were evaluated by three different CNNs, which had been previously trained by $H_2O_2$-induced senescence, CPT-induced senescence, and mixed $H_2O_2$- and CPT-induced senescence (Fig. 1e). The averaged classification accuracy was over 0.9, and the *F*1 score was also greater than 0.85 in every dataset (Fig. 1f, g and Supplementary Fig. 4a, b). Importantly, the CNN trained by $H_2O_2$-induced senescence recognised senescence not only in newly acquired $H_2O_2$-induced senescence datasets but also in CPT- and replication-induced senescence. Similarly, the CNN trained by CPT-induced senescence and CPT- and $H_2O_2$-induced senescence showed high performance in classifying senescence from controls under every condition (Fig. 1f, g and Supplementary Fig. 4a–f). AUCs were greater than 0.95 under every condition (Fig. 1h and Supplementary Fig. 4g, h), which supports a successful identification system for senescent cells.

Moreover, we examined whether the CNN can be applied to datasets obtained at another institution, Kyoto University. HUVECs were cultured and phase-contrast images were acquired at Kyoto University. The CNN was successfully trained on both the Keio (our institution) and Kyoto datasets with high performance (Supplementary Fig. 5a, b). We tested the performance of the CNN on Kyoto datasets, which were not used for training, and found that the CNN trained on the datasets from both institutes have a higher performance (Supplementary Fig. 5c). Importantly, the CNN also has a high performance with the Keio

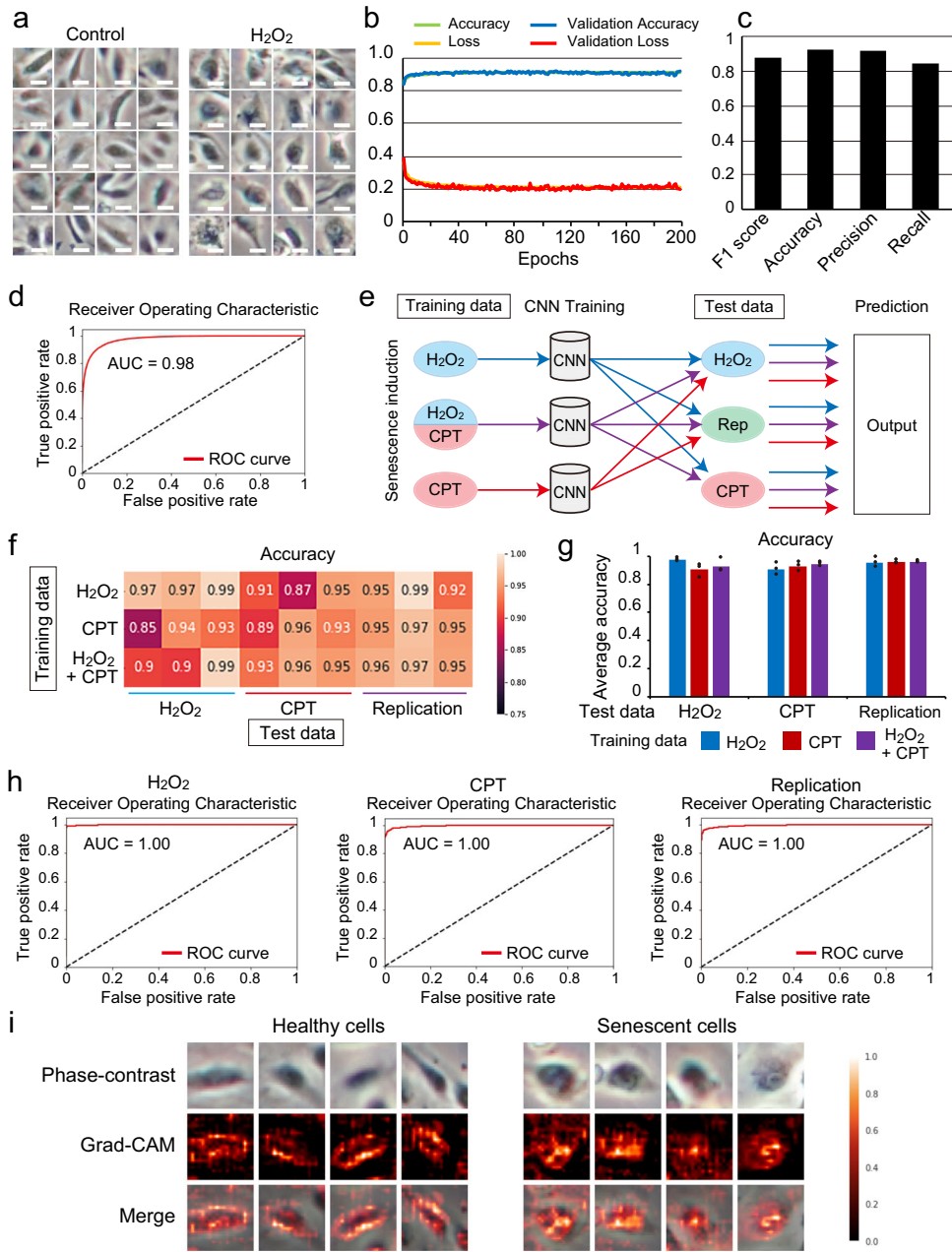

**Fig. 1 CNN training to classify control and senescent cells. a** Representative images of input images. Input images of control and $H_2O_2$-induced senescent cells were cropped from phase-contrast microscopy images at single-cell resolution by the OpenCV-based script. Scale bar, 7.1 μm. Data are representative of over three independent experiments. **b** Learning curve through the CNN training. Accuracy and loss in the training data and validation accuracy and validation loss in the validation data show the process of training. **c** Indexes: $F1$ score, accuracy, precision, and recall in the final setting of training. **d** AUC of the ROC curve in the final setting of training. **e** Protocol for the evaluation of CNN generalisability. Each CNN was trained by either the images of $H_2O_2$-induced senescent cells, the images of CPT-induced senescent cells, or the mixed images of $H_2O_2$- and CPT-induced senescent cells. Newly acquired data were used as test data. In the test data, cellular senescence was induced by $H_2O_2$, CPT, or replication. Test data were evaluated by three pre-trained CNNs. **f** A heatmap shows the accuracy of CNN prediction in each test dataset. Three independent experiments and evaluations were conducted for each senescence induction method. **g** Macro-averaged accuracy for each evaluation ($n = 3$ independent experiments). **h** AUC of the receiver operating characteristic (ROC) curve in the test data evaluated by CNNs, which were pre-trained by the data from $H_2O_2$-, CPT-, or replication-induced senescent cells. Data are representative of three independent experiments. **i** Grad-CAM shows an important region for the prediction of healthy or senescent cells. Data are representative of three independent experiments. CNN convolutional neural network, CPT camptothecin, Rep replication.

datasets, which suggests that the CNN trained on datasets from both institutes has higher generalisability. We also examined whether the CNN could classify senescence in other cell types. We used human diploid fibroblasts (HDFs), induced cellular senescence by $H_2O_2$ or CPT, cropped input datasets at single-cell resolution levels, and trained the CNN to classify them

(Supplementary Fig. 5d). The CNN was successfully trained (Supplementary Fig. 5e), and had a high performance in the test datasets (Supplementary Fig. 5f). Interestingly, the CNN trained on HUVEC-datasets was also able to classify healthy and senescent HDFs (Supplementary Fig. 5g). These results suggest that cellular senescence shows a unique morphologic characteristic, and a

morphology-based CNN system can reliably identify senescent cells.

To better understand where CNN could identify senescent cells, we visualised important regions for senescent cell prediction by gradient-weighted class activation mapping (Grad-CAM). Grad-CAM incorporates class-specific gradient information into the final CNN convolutional layer to visualise important image regions[14]. Grad-CAM indicated that the CNN identified healthy and senescent cells by recognising peripheral and heterogeneous intracellular images, respectively (Fig. 1i). This information could help understand the biological meaning of cellular morphology.

**Development of Deep-SeSMo**. The output of the trained CNN was a non-linear prediction with two values, control (0) or senescence (1), meaning that the CNN classified cells as senescent or control, and there was no intermediate state. A drug-screening index ideally requires a quantitative evaluation. To examine the relationship between cellular senescence and the strength of senescence-inducing stress, we acquired phase-contrast images with several $H_2O_2$ or CPT doses or passage numbers (Fig. 2a and Supplementary Fig. 6a, b). *P21* expression was correlated with $H_2O_2$ and CPT concentrations and passage numbers (Fig. 2b–d), indicating that cellular senescence could be induced quantitatively in a stress strength-dependent manner. In the last CNN layer, the softmax function calculates class probability with the image belonging to either senescence or non-senescence. We then focused on the senescence probability for quantitative assessment. Interestingly, the senescence probability output from the pre-trained CNN mostly showed 0 or 1 at the single-cell level (Fig. 2e and Supplementary Fig. 6c); however, the ratio of senescent cells and average senescence probability correlated with the degree of cellular senescence induction (Fig. 2f). Therefore, we proposed a "senescence score" based on the pre-trained CNN, optimised it for the classification problem, and defined the overall average output probability calculated by the pre-trained CNN as a quantitative senescence score. Importantly, the senescence scores strongly correlated with the $H_2O_2$ and CPT concentrations and passage numbers (Fig. 2g–i and Supplementary Fig. 6d, e). The Pearson correlation coefficient demonstrated a high linear correlation between the score and stressors in all combinations (Fig. 2j). The networks trained by both $H_2O_2$- and CPT-induced senescence showed a correlation coefficient over 0.9 under any stress-induced senescence, including replication stress (Fig. 2k). We termed this strategy for the calculation of a senescence score using CNN training Deep-SeSMo. Deep-SeSMo could calculate the senescence score for each phase-contrast image in only 0.08–0.1 ms (Supplementary Fig. 7a). A senescence score which was generated by the CNN trained on the datasets acquired at two institutes, Keio and Kyoto (Supplementary Fig. 7b, c), or the CNN trained on another cell type, HDFs (Supplementary Fig. 7d, e), also showed high performance.

**Anti-senescent drug screening**. To validate the performance of Deep-SeSMo, we first examined the effects of well-known anti-cellular senescence reagents such as nicotinamide mononucleotide (NMN), a key NAD + intemediate[15], and metformin, an AMPK activator[16]. NMN and metformin decreased the SA-β-gal-positive cell ratio, P21–P53 activation, and P16INK4a expression (Fig. 3a–c). Deep-SeSMo successfully assessed the effects of NMN and metformin (Fig. 3d, e). Senolytics are focused as potential therapeutic drugs for age-related diseases to induce apoptosis specifically in senescent cells[17]. We examined whether Deep-SeSMo could correctly assess the senolytic effect of ABT263. We mixed the young and old HUVECs, treated them with ABT263,

and analysed the cells using Deep-SeSMo. Deep-SeSMo could also correctly assessed the senolytic effect of ABT263 (Supplementary Fig. 7f, g). We then conducted drug screening to repress cellular senescence utilising Deep-SeSMo (Supplementary Fig 1a). A kinase inhibitor library was used to screen compounds that suppress cellular senescence induced by the three methods in HUVECs and to understand the mechanism underlying cellular senescence. Senescence scores were calculated by Deep-SeSMo and normalised by a control sample (Fig. 3f). We repeated the screening twice with three senescence induction methods (Supplementary Fig. 8a, b). The senescence score was converted into senescence score ranking for each evaluation. The surface plot of senescence score ranking for each drug clearly showed that several drugs could suppress senescence (Fig. 3g). Most drugs showed the strongest effect on senescence promotion. To identify potential drugs for senescence suppression, we ranked the compounds by calculating the median senescence score ranking in all evaluations and focused on the top four compounds, terreic acid, PD-98059, daidzein, and Y-27632·2HCl, as anti-senescent drugs (Fig. 3h). We also established a heatmap image of senescence score ranking to visualise the effects of every drug and determine an anti-senescence cluster, in which the senescence phenotype was prominently suppressed (Fig. 3i and Supplementary Fig. 8b). The top four compounds were also included in an anti-senescence cluster.

Terreic acid, a metabolite of *Aspergillus terreus*, possesses antibiotic properties[18] and is a quinone epoxide inhibitor of Bruton's tyrosine kinase (BTK)[19]. Interestingly, terreic acid can extend yeast life span, even though yeast does not express BTK[20]. PD-98059 is a selective inhibitor of mitogen-activated protein kinase, a kinase of the extracellular signal-regulated kinase, and suppresses cellular senescence[21,22]. Daidzein is an isoflavone in soybean that suppresses ageing phenotypes[23,24]. Y-27632·2HCl is an inhibitor of the Rho-associated coiled-coil-forming kinase (ROCK), a member of the serine/threonine kinases, which regulates cell proliferation, apoptosis, migration, metabolism, and senescence[25,26]. We tested whether the selected compounds suppressed cellular senescence, using conventional experiments. SA-β-gal activity analyses showed that the four compounds decreased cellular senescence (Fig. 4a, b). Western blotting also demonstrated that the four compounds suppressed P53–P21 axis activation and P16INK4a expression (Fig. 4c). *P21* expression was also reduced by all compounds with the three senescence induction methods (Supplementary Fig. 9a–d). We also examined the effects of four drugs (SC-514, TYRPHOSTIN51, Indirubin, and SU4312, which were determined as non-effective drugs by Deep-SeSMo analysis, with almost the same senescence score as the control) on the P53–P21 senescence axis. All four drugs showed almost no effects on the activation of the P53–P21 signalling pathway (Supplementary Fig. 9e). This evidence suggests that Deep-SeSMo was reliable and could be used for drug screening.

**Underlying mechanisms for cellular senescence suppression**. Finally, we examined the mechanism by which these compounds suppress the senescence phenotype. Global gene expression analysis by RNA sequencing was conducted using senescent endothelial cells treated with each of the four compounds. A heatmap showed the top 10 genes among the differentially expressed genes for all four compounds and the control (Fig. 4d and Supplementary Fig. 9f). Nuclear factor kappa B (NFκB) is an important transcription factor that induces inflammatory SASP[27]. Among the top 10 genes, three genes were associated with NFκB function: *TBL1XR1* (ref. [28]), an NFκB activator, was down-regulated, and *SIGIRR*[29] and *ASCC1* (ref. [30]), NFκB inhibitors,

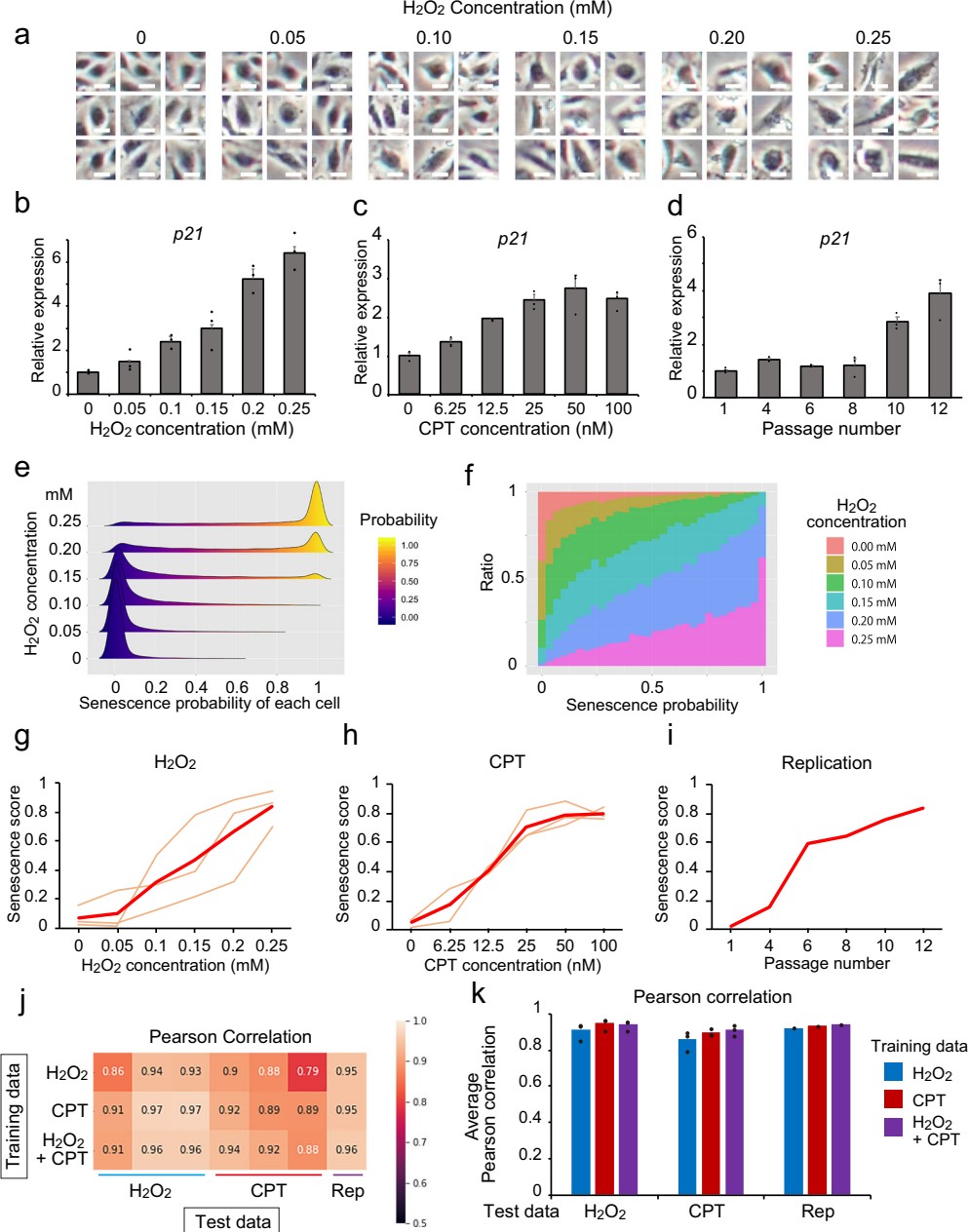

**Fig. 2 Development of Deep-SeSMo. a** Representative input images of HUVECs treated with various concentration of $H_2O_2$. Scale bar, 7.1 μm. Data are representative of three independent experiments. **b**–**d** qRT-PCR analysis to determine the mRNA expression of *P21* under various **b** concentrations of $H_2O_2$, **c** concentrations of CPT, and **d** passage numbers (*n* = 3 biological replicates). **e** Density plots of all senescence probabilities, which were outputs of CNN prediction, in HUVECs treated with various $H_2O_2$ concentrations. Data are representative of three independent experiments. **f** A rate graph shows the ratio of prediction for each senescence probability with various $H_2O_2$ concentrations. Data are representative of three independent experiments. **g**–**i** Senescence score calculated by CNNs trained by $H_2O_2$- and CPT-induced senescent HUVECs, in HUVECs treated with various $H_2O_2$ concentrations, CPT concentrations, and passage numbers. The thin line and bold line indicate each score and the average score, respectively. **j** A heatmap shows Pearson correlations for each set of test data. Three independent experiments of $H_2O_2$- and CPT-induced senescence and one experiment for replication-induced senescence, evaluated by three trained CNNs, were conducted. **k** Macro-averaged Pearson correlation, evaluated by three trained CNNs (*n* = 3 independent experiments for $H_2O_2$- and CPT-induced test data, *n* = 1 for rep-induced test data). CNN convolutional neural network, CPT camptothecin, Rep Replication. Data are shown as mean ± s.e.m.

were upregulated (Fig. 4d). Gene set enrichment analysis (GSEA) showed that genes related to the inflammatory response and NFκB signalling were negatively enriched in the four compounds (Fig. 4e, f). These results indicate that the four compounds not only suppress the senescence pathway but also the inflammatory phenotype. Terreic acid, the top compound, is a BTK inhibitor[19], but the expression of *BTK* was faint in HUVECs (Supplementary

Fig. 9g). Thus, the mechanism of terreic acid activity in HUVECs remains unclear. Gene ontology (GO) analysis demonstrated that terreic acid uniquely upregulates genes related to the positive regulation of ATPase activity in the mitochondria (Fig. 4g and Supplementary Fig. 9h, i). In senescent cells, mitochondrial function and ATP production via oxidative phosphorylation (OXPHOS) impairments have been observed[31]. RNA sequence

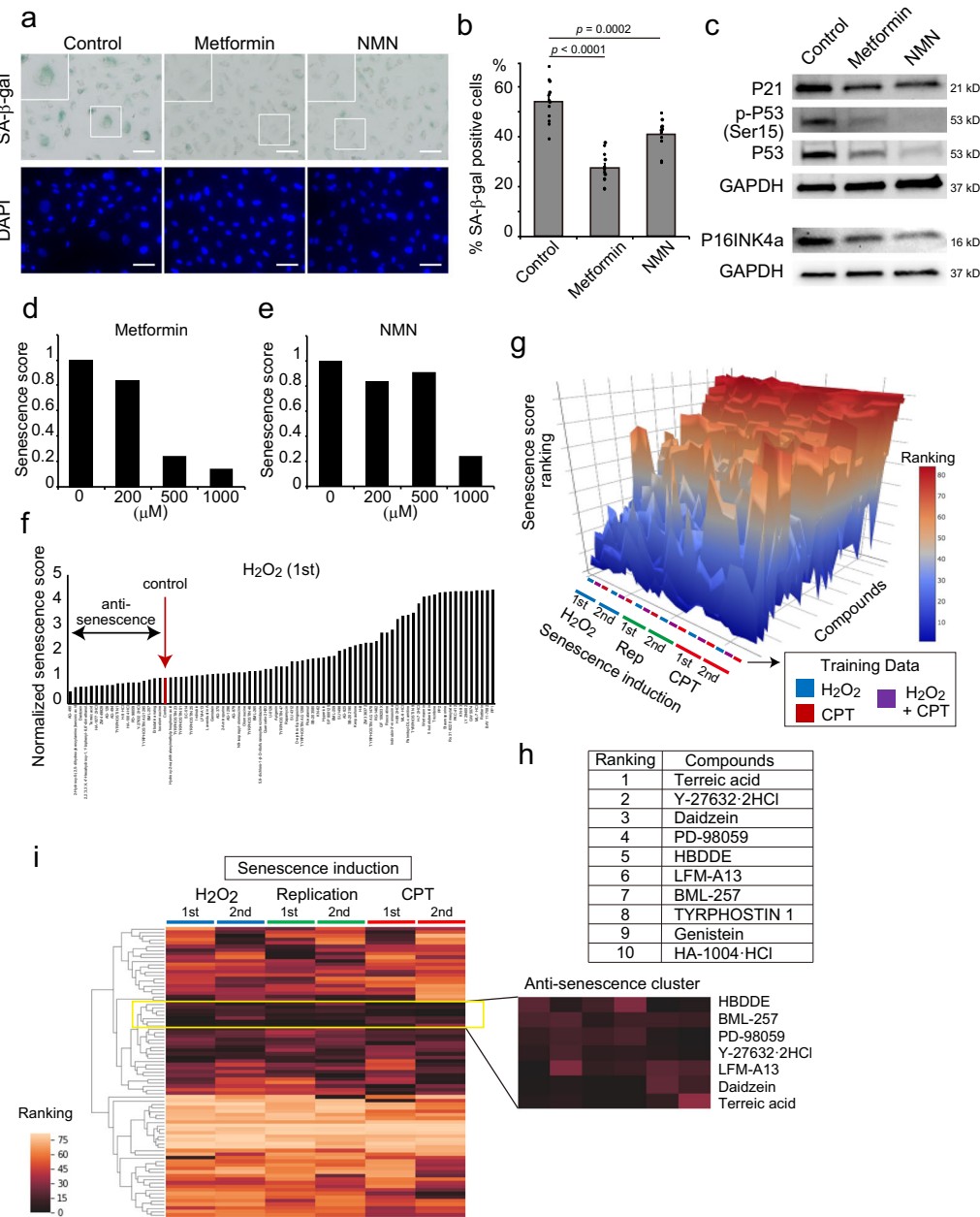

**Fig. 3 Drug screening using Deep-SeSMo. a** Representative images of SA-β-gal activity in senescent HUVECs treated with metformin or NMN. Scale bar, 100 μm. DAPI indicates the cell nuclei. Data are representative of two independent experiments. **b** Percentage of SA-β-gal-positive cells per total cells in senescent HUVECs treated with metformin or NMN ($n = 12$ images over two independent experiments). **c** Western blotting of P21, P53, Ser15 phosphorylation of P53, and P16INK4a in senescent HUVECs treated with metformin or NMN. GAPDH was used as an internal control. Data are representative of two independent experiments. **d**, **e** Senescence score calculated by Deep-SeSMo of senescent HUVECs treated with **d** metformin and **e** NMN. Data are representative of two independent experiments. **f** Eighty kinase inhibitors were added to HUVECs, and cellular senescence was induced by $H_2O_2$. The senescence score was calculated by Deep-SeSMo and normalised to a control score. **g** Senescence score ranking for 80 kinase inhibitors and control. Rankings were calculated for three stressors with two replications and evaluated by three pre-trained CNNs. A surface plot shows senescence score ranking sorted by the median value of senescence score ranking. **h** Top ten compounds detected by Deep-SeSMo. **i** A heatmap demonstrates a senescence score ranking for each condition evaluated by CNNs trained by $H_2O_2$- and CPT-induced senescent HUVECs. The right map shows an anti-senescence cluster that strongly suppresses senescence. CPT camptothecin, Rep replication, NMN nicotinamide mononucleotide. Data are shown as mean ± s.e.m. $p$ values by two-sided Student's $t$-test.

results suggest that terreic acid would maintain mitochondrial function under stress conditions (Fig. 4h, i). The inflammatory response and NFκB signalling were also attenuated by terreic acid treatment (Supplementary Fig. 9j, k). Terreic acid could be a drug against senescence and age-related diseases.

In conclusion, we established a drug screening method by constructing a rapid, accurate, morphology-based CNN system to identify senescent cells with Deep-SeSMo and identified potential drugs to suppress senescence.

## Discussion

In this study, we developed a drug-screening system for cellular senescence using a pre-trained CNN optimised by the overall average value of output senescence probability. Moreover,

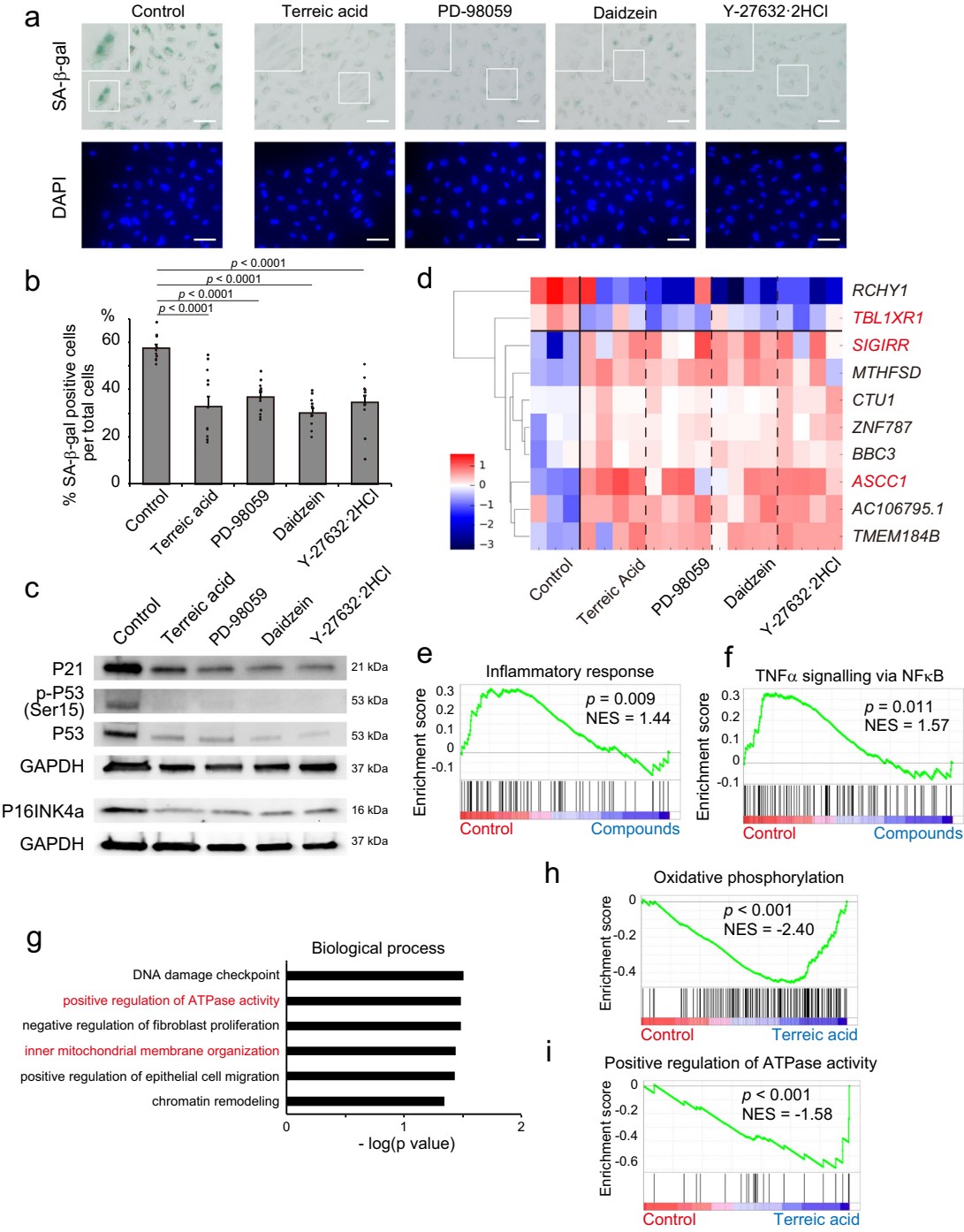

**Fig. 4 Anti-senescent effects of the top four compounds. a** Representative images of SA-β-gal activity in senescent HUVECs treated with the top four compounds: terreic acid, PD-98059, daidzein, and Y-27632. Scale bar: 100 μm. Data are representative of two independent experiments. **b** Percentage of SA-β-gal-positive cells ($n = 12$ images over two independent experiments). **c** Western blotting of P21, P53, Ser15 phosphorylation of P53, and P16INK4a. The top four compounds were added to HUVECs, and cellular senescence was induced by $H_2O_2$. GAPDH was used as an internal control. Data are representative of two independent experiments. **d** A heatmap shows the top 10 genes selected by differential gene expression analysis among all four compounds and control. The genes designated in red are associated with NFκB function. **e** GSEA of genes associated with the inflammatory response. **f** GSEA of genes associated with TNFα signalling via NFκB. **g** GO analysis, categorised as biological process, of genes upregulated in HUVECs treated with terreic acid compared with control. **h, i** GSEA of genes associated with **h** positive regulation of ATPase activity and **i** oxidative phosphorylation. NES normalised enrichment score. Data are shown as mean ± s.e.m. $p$ values by two-sided Student's $t$-test.

utilising a non-biased method, we identified four compounds, terreic acid, PD-98059, daidzein, and Y-27632·2HCl, which showed anti-senescent and anti-inflammatory effects. Drug development is facilitated by sophisticated screening systems. A human cannot reliably identify cellular status by observing

cellular morphology. However, cellular morphology can be a specific marker for cell type and pathological conditions because of specific morphological dynamics, including changes in protein expression and structure, and chromatin structure. In recent years, CNN has become a standard method to assess morphology.

CNN is most suitable for classification tasks; however, it is unclear whether quantitative analyses by CNN would be effective in the biological field. The concept of our strategy was simple; the overall average of output probability calculated by a pre-trained CNN was applied to the quantitative senescence score. Interestingly, a histogram of senescence probability showed that healthy cells would digitally transit into a senescent state, with a few cases of cells being in an intermediate state (Fig. 2e and Supplementary Fig. 6c). This suggests that cellular senescence would be induced digitally, and a less intermediate state might be observed during physiological ageing. Under intermediate stress conditions, the senescence probability is bipolarized, suggesting that senescence thresholds differ among cells. It would be interesting to elucidate the biological mechanism underlying the digital transition and threshold of cellular senescence. Although the CNN showed high performance, there were still mispredictions. When we output the false decision images (Supplementary Fig. 3g), the morphological appearance of false-positive images was similar to that of true-positive images, and false-negative images were similar to true-negative images. These suggest that a very small proportion of senescent cells exist in healthy conditions, and a very small proportion of healthy cells exist in senescence-inducing conditions, even though we paid full attention to the preparation of healthy or senescent cells. However, in our current analysis, incorrect predictions of the CNN were very rare; therefore, we believe that any incorrect predictions would have very little effect on the computation of the senescence score.

In this study, we identified several compounds repressing the senescence phenotype in vitro. A global transcriptome analysis indicated that these compounds have anti-inflammatory effects, via suppression of NFκB signalling. NFκB plays a central role in inflammation and the appearance of SASP[32], suggesting that these compounds could be strong candidates for a treatment against age-related diseases. Interestingly, the anti-senescent effects of terreic acid, which was the top candidate in our screening, have not been reported previously. Terreic acid is a BTK inhibitor, but BTK is not expressed in endothelial cells. Our results indicate that terreic acid improves mitochondrial function and ATP production via OXPHOS. Interestingly, a drug screening for life-extending compounds in yeast revealed that terreic acid can extend the mean replicative life span by 15%[20]. Its precise mechanism of senescence suppression should be clarified by proteome and metabolome analyses and validation of its effect in animal models. Cellular senescence has a pivotal role in age-related diseases such as diabetes, heart failure, atherosclerosis, and cancer; therefore, it would be interesting to examine the effects of the identified compounds against these diseases.

Diseased cells show specific morphology in several pathological conditions, although a human cannot identify the differences. A CNN-based approach contributes to the establishment of a non-biased method to identify morphological differences in research and drug screening. We developed a quantitative scoring system that evaluates cellular status by pre-trained CNN. Deep-SeSMo may be applicable for drug screening in other diseases and as a landmark system for drug discovery.

## Methods

**Cell culture**. HUVECs (KURABO) were cultured on gelatin-coated dishes with HuMedia-EG2 medium (KURABO). In total, 100,000 cells (HUVECs) per well were plated one day before senescence induction. The HDF cell line, TIG-114, was purchased from the Japanese Collection of Research Bioresources (JCRB) Cell Bank. HDFs were cultured on gelatin-coated dishes with Eagle's minimum essential medium with 10% FBS. HDFs were plated at 50,000 cells per well, one day before senescence induction.

**Microscopic imaging**. Phase-contrast images of the control and senescent HUVECs were acquired using inverted microscopy (Olympus). We prepared the images of senescent HUVECs for training datasets; senescence was induced by either $H_2O_2$ or CPT. For training datasets, 10 images were acquired from over four independent experiments. For the test datasets, we induced cellular senescence in HUVECs by three methods: $H_2O_2$, CPT, and replication. Five images were acquired from three independent experiments for test data. Each image was saved as a 2776 × 2074 px RGB image in Tiff format.

**RNA isolation and reverse transcriptase PCR**. Total RNA was collected using TRIZOL (Thermo Fisher) and the ReliaPrep RNA cell miniprep system (Promega). cDNAs were prepared utilising the ReverTra Ace qPCR RT master mix with gDNA Remover (Toyobo). The following primers were used: human β-actin forward (GCAAAGACCTGTACGCCAAC) and reverse (AGTACTTGCGCTCAGGA GGA) and human *P21* forward (TCAGGGTCGAAAACGGCG) and reverse (AAGATCAGCCGGCGTTTGGA).

**SA-β-Gal staining**. Control and senescent HUVECs were washed twice with PBS, fixed with 4% paraformaldehyde (WAKO) for 10 min at room temperature, washed twice with PBS again, and incubated in 1 ml of SA-β-gal staining solution containing 100 mM $K_4[Fe(N)_6]3H_2O$, 100 mM $K_3[Fe(CN)_6]$, 1 M $MgCl_2$, and 20 mg ml$^{-1}$ X gal overnight at 37 °C. The nuclei were stained by 4′,6-diamidino-2-phenylindole for 10 min at room temperature. SA-β-gal activity was detected by bright field images using a fluorescence laser microscope (BZ-9000, KEYENCE).

**Western blotting**. We extracted protein utilising RIPA buffer (Nacalai tesque) and measured protein concentration by Pierce BCA protein assay kit (Thermo Fisher Scientific). Proteins were electrically separated by Mini-PROTEAN TGX gels (Bio-Rad, 4–15%) and transferred to nitrocellulose membranes with 0.2 μm pore size by dry blotting (Invitrogen). The membranes were incubated with a primary antibody for P21 (#2947, 12D1, Cell Signaling), P53 (#ab1101, DO-1, Abcam), phospho-P53 (#9284, Cell Signaling), P16INK4a (#ab108349, Abcam), or GAPDH (#2118, 14C10, Cell Signaling) overnight at 4 °C and with a horseradish peroxidase-conjugated secondary antibody (anti-mouse HRP 1:2000 or anti-rabbit HRP 1:2000) for 1 h at room temperature. Chemi-Lumi One L or Chemi-Lumi One Super (Nacalai tesque) was used for the visualisation of immunoreactive bands. Images were acquired utilising an Image Reader LAS-3000 (FUJIFILM).

**GPU server and analysis environment**. We used GPGPU server, which has two CPUs: Xeon 4-Core E5-2637V4 3.5 GHz, 128 GB CPU memory, and two GPUs, GeForce GTX1080Ti GDDR5 11GB (NVIDIA, Santa Clara, CA, USA). We programmed all scripts on Nvidia-docker system with Ubuntu 16.04, CUDA 8.0, cuDNN 6.0, Anaconda 3 4.4.0, Python 3.5, Tensorflow 1.4.0, and KERAS 2.1.2.

**Senescence induction**. We induced cellular senescence in HUVECs by three approaches: $H_2O_2$, CPT, and serial passages. For the CNN training, HUVECs were exposed to 0.25 mM $H_2O_2$ in serum-free EGM2 medium (Lonza) without antibiotics and ascorbic acid for 4 days, or 100 nM CPT in serum-free EGM2 medium (Lonza) without antibiotics and ascorbic acid for 2 days, to obtain senescent HUVECs. HUVECs passaged over 10 times and cultured in serum-free EGM2 medium (Lonza) without antibiotics and ascorbic acid for 5 days were used as old HUVECs produced by replication stress. For control samples, HUVECs were cultured for 4 days for the $H_2O_2$ control and for 2 days for the CPT control. The passage number for controls of old HUVECs was lower than three. Cellular senescence in HDFs was induced by 0.02 mM $H_2O_2$ for 4 days, or 200 nM CPT for 3 days. For control, HDFs were cultured with 10% FBS EMEM for 4 days for the $H_2O_2$ control and for 3 days for the CPT control. For the test datasets, cellular senescence in the HUVECs and HDFs was induced using the same conditions as the training datasets. To calculate the senescence score, 0, 0.05, 0.1, 0.15, 0.2, and 0.25 mM $H_2O_2$; 0, 6.25, 12.5, 25, 50, and 100 nM CPT; and 1, 4, 6, 8, 10, and 12 passages were used for HUVECs. The senescence score for HDFs was calculated using 0, 0.005, 0.01, 0.015, 0.02, and 0.025 mM $H_2O_2$; 0, 6.25, 12.5, 25, 50, and 100 nM CPT. For the drug screening or drug assessment in HUVECs, 0.15 mM $H_2O_2$ was added for 4 days or 25 nM CPT for 2 days, with simultaneous application of the test compounds. For the replicative stress screens, moderately senescent HUVECs were incubated with compounds for 5 days. For RNA sequence analysis, we used moderately senescent HUVECs. As a control for the drug treatments, the same concentration of dimethyl sulfoxide (DMSO) was used alongside senescence induction.

**Automated single-cell cropping**. To generate input datasets for training, validation, and drug screening, we cropped phase-contrast images at single-cell resolution. The acquired images were binarized by a predetermined threshold, and the cell locations were identified by black particles. The threshold value for cell size was determined, and we confirmed that the cells could be correctly identified. We also defined the noise particles as being smaller than the cells. Thus, cropped images under the defined size were automatically eliminated for further analysis. The positions of centre gravity of the cells were identified using the OpenCV package. Next, 50 × 50 px input datasets were automatically cropped from the original phase-contrast images at locations determined by the centre of cell gravity in

binarized images. Input datasets were converted to numpy array. All programmes were written using Python 3.

**Training by a deep neural network**. We used a convolutional neural network for training. The network consisted of four convolution layers, two max pooling layers, and two fully connected layers. Each convolutional network was connected to rectified linear units for activation. The final layer was connected with the softmax function to calculate the probability of classification. To avoid overfitting, dropout techniques were used following the layer of first and second Max pooling, and the first dense layer. The dropout rate was 0.5. We utilised mini-batch training with stochastic gradient descent method, learning rate 0.032, and cross-entropy error as loss function. Weights were initialised using the Grolot uniform value. We normalised input images as follows:

$$Y = ((X/255) - 0.5) \times 2. \tag{1}$$

$Y$: value of normalised images and $X$: value of original images.

The max value of normalised images were "1" and the min value of normalised images were "−1". To increase input data on the computer, we used data augmentation method, rotation, width shift range, height shift range, horizontal flip, and vertical flip. The trained networks can output whether the input data represent control cells or senescent cells. We constructed three trained networks, which classified control versus $H_2O_2$-induced senescence, control versus CPT-induced senescence, and control versus both $H_2O_2$- and CPT-induced senescence. For datasets of training, we obtained 10 phase-contrast images each from over four independent inductions of senescence. The trained network performance was accessed by accuracy, loss value, recall, precision, $F1$ score, and AUC of the ROC curve. For the training of senescent and healthy HDFs, we obtained 10 phase-contrast images each from three independent inductions of senescence using both $H_2O_2$ and CPT.

**Training by traditional machine learning methods**. For training using feature-based traditional machine learning, we used the same datasets of healthy and senescent HUVECs that were used for CNN training. HOG was used for a feature descriptor of images, and 2916 length feature vectors were output in each image. Logistic regression, Random forest, and linear kernel of Support vector classifier were used to classify healthy and senescent HUVECs. The performance was validated by accuracy, $F1$ score, and AUC of the ROC curve, and compared with CNN performance.

**Evaluation of the trained networks**. The performance of trained networks was validated with newly acquired test datasets. We prepared three datasets: $H_2O_2$-, CPT-, and replication-induced senescence. We obtained five phase-contrast images from each of the three independent senescence inductions. Each dataset was classified using three networks; thus, in total, 27 results were obtained for evaluation. Accuracy, recall, precision, $F1$ score, and AUC of ROC curve were used for network evaluation. Grad-CAM was used to visualise important regions of healthy and senescent cells. Detailed code for Grad-CAM can be available at public GitHub repository [https://github.com/Dai-Kusumoto/Deep-SeSMo].

**Senescence Scoring System (Deep-SeSMo)**. We calculated the senescence score using the probability of senescence, which represents the outputs of trained classification networks to detect senescent cells, for each cell at the single-cell resolution. The neural network was able to calculate the probability class of input data, when the softmax function was used in the output layer:

$$\text{softmax}: y_i = \frac{e^{x_i}}{\sum_{j=1}^{N} e^{x_j}}. \tag{2}$$

The average probabilities of belonging to the senescence class in all input datasets were used for the senescence score:

$$\text{senescence score} = \frac{\sum_{k=1}^{m} y_{1k}}{m} \tag{3}$$

($m$: the number of input datasets and $y_1$: the probability of belonging to the senescence class in each input dataset). The correlation between senescence score and stress strength was quantified by Pearson correlation. Stress strengths were obtained by the concentrations of $H_2O_2$ or CPT, or several passage numbers. For drug assessment, the senescence score was normalised by the control sample. For evaluation using several samples, each senescence score was converted to ranking of senescence score for each evaluation. To prioritise drugs, the median of senescence score ranking was used. For the evaluation of Deep-SeSMo, 0, 200, 500, and 1000 μM metformin and 0, 200, 500, and 1000 μM NMN were added to HUVECs for 4 days, and the senescence score was calculated by Deep-SeSMo. Visualisation of senescence probability was performed using the ggplot2 package.

**CNN validation at another institute**. To validate the CNN performance at another institute, HUVECs were cultured and cellular senescence was induced by either $H_2O_2$ or CPT at Kyoto University as per our protocol. Induction of cellular senescence was repeated twice with both $H_2O_2$ and CPT conditions. Then, phase-contrast images were acquired and input datasets were generated by the defined

method. The number of obtained datasets was 57,115 for healthy and 25,471 for senescent HUVECs. We mixed both the Kyoto University and Keio University (our institute) datasets, obtained 281,842 healthy and 140,061 senescent cell datasets, and trained the CNN to classify healthy and senescent HUVECs. The performance of the CNN (Keio + Kyoto) was compared with the CNN trained only on datasets obtained at our institute (Keio). To assess the external validation of the CNN (Keio + Kyoto), we prepared the datasets which were obtained in the first senescence induction at Kyoto University (33,041 for healthy and 14,425 for senescence) and mixed them with a random arrangement of the same number of datasets obtained at our institute. We then trained the CNN and validated the datasets which were obtained in the second senescence induction at Kyoto University (24,074 for healthy and 12,046 for senescence). Test datasets from our institute (Keio) were used as well, as shown in Fig. 1f–h. For validation of senescent score, the CNN trained on the all mixed Keio and Kyoto datasets was used, and HUVECs with 0, 0.05, 0.1, 0.15, and 0.2 mM $H_2O_2$; or 0, 12.5, 50, and 200 nM CPT were used to calculate the senescence score.

**Senolytics treatment**. We mixed young (100,000 cells per well) and old HUVECs (200,000 cells per well) one day before drug treatment. 0.25, 0.5, 1, and 2.5 μM of ABT263 (Adoop) were added to the HUVECs for 3 days. Five phase-contrast images were acquired in each condition, including before drug treatment, and senescence score was calculated using Deep-SeSMo.

**Drug screening**. We performed drug screening by utilising Deep-SeSMo. We tested 80 compounds using a kinase inhibitor library (SCREEN-WELL® Compound Library, Enzo). Ten micromolar drugs were added with three senescence induction methods: $H_2O_2$, CPT, and replication. The same concentration of DMSO was used for control samples. The senescence score was calculated utilising three networks trained by $H_2O_2$, CPT, and $H_2O_2$ and CPT. Subsequently, the senescence score was converted to ranking of senescence score in each analysis. To detect an anti-senescent cluster, a heatmap was drawn using all samples with validation by $H_2O_2$ and CPT-trained networks by the seaborn package. To calculate the anti-senescent ranking of drugs, the median of the senescence score ranking was sorted. A visualisation of the senescence score ranking was performed by the Plotly package. For the evaluation of anti-senescence effects, four compounds, 10 nM terreic acid, 500 nM Y-27632·2HCl, 5 μM daidzein, and 100 nM PD-98059, were used. Four compounds from the kinase inhibitor panel (SC-514, TYRPHOSTIN51, Indirubin, and SU4312) were selected for the validation of non-effective drugs, which displayed almost the same senescent score as the control by Deep-SeSMo analysis.

**Evaluation of network performance**. Network performance was evaluated by accuracy, precision, recall, and $F1$ score, and the AUC of ROC curve. Accuracy is the ratio of correct predictions to all predictions. Precision is the hitting ratio of positive predictions. Recall is the sensitivity of prediction.

$$\text{accuracy} = \frac{TP + TN}{TP + FP + TN + FP}, \tag{4}$$

$$\text{precision} = \frac{TP}{TP + FP}, \tag{5}$$

$$\text{recall} = \frac{TP}{TP + FN}. \tag{6}$$

$F1$ score is the combination of recall and precision:

$$F1 \text{ score} = \frac{2 \text{ Recall} \times \text{Precision}}{\text{Recall} + \text{Precision}}. \tag{7}$$

The ROC curve is the plot of true-positive rate against false-positive rate for all possible thresholds.

**RNA sequence analysis**. We added four compounds: 10 nM terreic acid, 500 nM Y-27632·2HCl, 5 μM daidzein, and 100 nM PD-98059 to HUVECs for 4 days, and mRNA was extracted as described above. The library for sequencing was prepared according to the manufacturer's protocol (NEBNext® Ultra™ II RNA Library Prep Kit for Illumina®). Next, $2 \times 150$ bp pair-end (PE) sequencing was carried out by Illumina Hiseq. Sequencing data were converted into fastq format using the bcl2fastq software. The sequence quality was checked by the FastQC software, and we eliminated adapter sequences by Trimmomatic. We processed the sequence data and got FPKM using the HISAT2-StringTie-Ballgown pipeline. Differential expressed genes between control and all four compounds or control and terreic acid were calculated using Ballgown packages. Heatmap showed a distance from the average logFC of the median control value and median values of all four compounds. GSEA was performed using genesets in Molecular Signatures Database. Gene ontology analysis of upregulated genes (fold change >1.5) in terreic acid-treated HUVECs compared with control was carried out by the Database for Annotation, Visualisation, and Integrated Discovery (DAVID).

**Reporting summary**. Further information on research design is available in the Nature Research Reporting Summary linked to this article.

## Data availability
Source data for figures are provided with the paper. For RNA sequence data in Fig. 4 and Supplementary Fig. 9, raw data have been deposited in DDBJ Sequence Read Archive (DRA) with the accession code "DRA010959".

## Code availability
Custom code for CNN training, validation, senescence scoring based on Deep-SeSMo can be available at public GitHub repository [https://github.com/Dai-Kusumoto/Deep-SeSMo].

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

## Acknowledgements
We thank all the members of our laboratory for their assistance. This research was supported by AMED under Grant Number JP18bm0404026, Grants-in-Aid for Scientific Research (JSPS KAKENHI, Grant Number 19K08549), Keio Gijuku Academic Development Funds, Grant for Basic Research of the Japanese Circulation Society (2020), and the Keio University Medical Science Fund.

## Author contributions
D.K. and S.Y. designed the experiments. D.K., T.S., H.S., A.K.,T.K., M.K., S.I., J.K., and H.H. collected the data. D.K. and H.S. analysed the data. K.F. supervised the research. D.K. and S.Y. wrote the article.

## Competing interests
K.F. is a Founding Scientist and funded by the SAB of Heartseed Co. Ltd. D.K., T.S., H.S., A.K., T.K., M.K., S.I., J.K., H.H., and S.Y. declare no competing interests.
