## [Peer Review File · Nature Communications]

Reviewers' Comments:

Reviewer #1:

Remarks to the Author:

The manuscript presents a deep learning-based framework for anti-senescent drug screening. Specifically, it first trains a convolutional neural network (CNN) to identify senescent cells from phase-contrast microscopy images. Then, it calculates a senescence score using the predictions of the pre-trained CNN. Finally, it applies this scoring mechanism to drug screening and identifies some novel potential drugs to suppress senescence.

Detailed comments:

1. In the first step of the proposed method, it is a straightforward application of CNNs to image classification. Additionally, it simply uses the predictions of a classifier (here a CNN) for senescence scoring and anti-senescent drug screening. It is not clear why a deep learning model instead of other traditional machine learning (ML) classifiers is needed in this study. Compared with CNNs, other traditional ML methods typically require less training data and computing resources. It would be convincing to compare the CNN with other simple ML models such as SVM and random forests.
2. It is not clear how to calculate a quantitative senescence score based on the CNN predictions. For each input, the CNN outputs the probability of the input belonging to the i -th class as: $y_i = \exp(x_i) / \sum_j (\exp(x_j))$, as shown in Line 5 at Page 24. However, what does " $\sum_i (y_i) / n$ " mean in Line 7 at Page 24? What does n present? In addition, the senescence score is calculated as the average probabilities over all cells in a single input image or in the entire test set?
3. Can the proposed method be trained on a dataset but be used on another dataset that is from a different institution?
4. An intensity thresholding method is used to crop cell patches for CNN training. Does the threshold value change for different images? Is it possible that this thresholding method generates noisy patches such that there are no cells inside these patches or the cells are not centered? If so, how does the proposed method deal with these cases? How will it perform in the presence of noisy data? How much noises it can handle?
5. Is it possible to directly link the discriminative regions identified by the Grad-CAM to the senescence score computation and thus improve the interpretability of Deep-SeSMo? How can these important regions be used to analyze the effects of anti-cellular senescence reagents or understand cellular senescence suppression (if applicable)?
6. How will incorrect predictions of the CNN affect the computation of the senescence score? It will be helpful to analyze/discuss the failures and the limits of the proposed method.
7. Which layer of the CNN uses dropout? What is the dropout rate?
8. Are the input images normalized before feeding them to the CNN? If so, how?

Reviewer #2:

Remarks to the Author:

In their recent manuscript, Kusumoto et al. present a novel CNN (Deep-SeSMo) capable of automated image analysis to discriminate between senescent HUVECs and control endothelial cells under multiple senescence-inducing conditions (replicative stress, camptothecin, and peroxide). The authors validate Deep-SeSMo using several senescence-suppressing compounds before

applying their technique to screen an 80-member kinase inhibitor library for potential novel senescence-suppressing molecules. These results are of great potential interest, particularly considering the growing interest in developing senolytic or senomorphic compounds for treatment of age-related diseases. However, I have several concerns that, should they be addressed, would greatly strengthen the manuscript.

1.) Clarification of the methods for drug treatments and screening. It is implied but not explicitly stated in the Methods or main text that senescent HUVECs were generated prior to treatment with either the candidate kinase inhibitor library and the hits from CNN screening. It is essential to clarify how senescent HUVECs were generated and when in the timeline of senescence induction the compounds were applied. It would also be important to test at least 4 compounds from the kinase inhibitor panel that did not show senomorphic activity based on Deep-SeSMo and show that these do not have an effect on p53-p21 axis activity. Furthermore, the authors do not examine effects on proliferative arrest or p16 expression, other major hallmarks of cellular senescence, either in their senescent cell cultures with or without senomorphic treatment used as the training datasets or the 4 kinase inhibitor hits identified by Deep-SeSMo.

2.) All presented validation and screening focuses on "senomorphic" compounds and in HUVECs specifically. Essentially, the authors focus on determining whether Deep-SeSMo is capable of detecting senescence-suppressing effects of known senomorphic compounds (such as metformin) and identifying novel compounds with similar effects. While this is indeed of significant value, a great deal of drug discovery focus within the senescence field is on senolytic (senescent cell killing) drug identification. It would be of considerable value if the authors could apply their CNN to determine whether known senolytics, such as ABT263 (navitoclax), are capable of depleting senescent HUVECs from a mixed culture of senescent and control cells.

Additionally, all observations based on Deep-SeSMo are limited to HUVECs in monolayer culture. While this is certainly important proof of principle that this image analysis approach is viable for drug screening for senomorphics in senescent cell culture, it would be encouraging to know that a CNN developed on similar principles would be able to, at least, discriminate between senescent and control cells of a non-endothelial origin, such as human diploid fibroblasts or mouse embryonic fibroblasts. I suggest these cell types because they also reliably respond to peroxide, DNA-damaging agents, and extended passaging with the senescence response.

In summary, the authors have produced a potentially very useful approach to automated identification of senescent cells at the single cell level using morphology alone. However, this tool was only validated using HUVECs; several other classic markers of cellular senescence (such as p16 and proliferative status) were not assessed; and, non-senomorphic hits from the kinase inhibitor screen were not checked to show that they do not alter molecular parameters of senescence in HUVECs. These are the most important limitations of the current study and addressing them would greatly increase confidence that Deep-SeSMo, or a related CNN, would have broad utility in identifying senomorphic compounds irrespective of cell type. Additionally, testing to see whether Deep SeSMo can also detect senescent cell killing (senolytic) compounds would enhance confidence that Deep SeSMo would be useful within the current senescence/drug discovery initiatives.

Reviewer #3:

Remarks to the Author:

In this manuscript, Kusumoto et al., describe anti-senescent drug screening by deep learning-based morphology senescence scoring. Specifically, authors developed a new convolutional neural networks (CNN)- based technique of senescence scoring. The technique termed as "Deep Learning-Based Senescence Scoring System" (Deep-SeSMo) appears to have a great potential of advancing drug -discovery, particularly in the area of ageing and cancer. The authors established

this innovative method based on senescent phenotype observed in endothelial cells. Furthermore, authors successfully identified four novel anti-senescent drugs using this method. The utility of this innovative approach is validated further analysis of the compounds, which appear to inhibit a well-known inflammatory response pathway, otherwise known as "Senescence-Associated Secretory Phenotype" (SASP), a hallmark of the senescent cells. It is a well-done study, and a well-written manuscript. It should be of very interest to ageing and cancer researchers as well as pharma companies. Authors have used multiple types of stress agents that are known to induce premature senescence in human umbilical vein endothelial cells (HUVECs). The senescent phenotype scored by CNN based scoring methods has been verified by known markers of senescence such as SA- β -gal. The minor concern is about the wider applicability of the method to cell types other than HUVECs. The morphological features of senescence may vary in different cell types. Authors could further validate Deep-SeSMo using a different cell type such as a strain of human diploid fibroblast (HDF). The other specific comments are as following-

1. Results- Page 5 line 6, senescent cells are resistant to apoptosis, it is incorrect to say p21-p53 pathway induces apoptosis in senescent cells; hence remove apoptosis from sentence "p21-P53 pathway---- cell cycle arrest and apoptosis".
2. Figure 2J and K- the graph indicates three replicates each for H₂O₂, CPT and Rep, however only one replicate is shown for Rep in heat map, and only one replicate indicated for Rep in the figure legend.
3. Fig. 3 and 4- Please indicate how was senescence induced in control cells?
4. Figure 3C, Fig. 4C- it would be good to include p16INK4a in these Westerns.

Point to point response

Reviewer #1:

The manuscript presents a deep learning-based framework for anti-senescent drug screening. Specifically, it first trains a convolutional neural network (CNN) to identify senescent cells from phase-contrast microscopy images. Then, it calculates a senescence score using the predictions of the pre-trained CNN. Finally, it applies this scoring mechanism to drug screening and identifies some novel potential drugs to suppress senescence.

Thank you very much for your careful reviewing and valuable comments. As suggested, we have performed additional experiments and analyses to improve the quality of our study, and have revised the manuscript accordingly.

Detailed comments:

1. In the first step of the proposed method, it is a straightforward application of CNNs to image classification. Additionally, it simply uses the predictions of a classifier (here a CNN) for senescence scoring and anti-senescent drug screening. It is not clear why a deep learning model instead of other traditional machine learning (ML) classifiers is needed in this study. Compared with CNNs, other traditional ML methods typically require less training data and computing resources. It would be convincing to compare the CNN with other simple ML models such as SVM and random forests.

Thank you for your comment. As you pointed out, it is important and interesting to compare convolutional neural networks and classical machine learning models, and we therefore tested our CNN against the Support Vector Machine (SVM), Random forests, and logistic regression models. To analyze cellular images using classical machine learning models, we first had to extract features of images to create input datasets. Several methods have been developed for feature descriptors. Of these, we used Histograms of Oriented Gradients (HOG), which is one of the most common methods, as the feature descriptor of cellular images. We used the microscopic images that were used for CNN training in Figure 1a. The number of images that we used here was 92,242 for H₂O₂-induced senescence, 41,207 for H₂O₂ control, 134,097 for CPT-induced senescence, and 64,535 for CPT control. We trained the traditional machine learning models to classify control and senescent cells, and compared them with CNN. The accuracy and F1 score of traditional machine learning classifiers were lower than that of the CNN, and we concluded that the CNN is the most suitable method to deal with classification of senescent cells (Extended Data Fig 3f). These results were similar to those of a previous report

(Buggenthin F, et al. *Nature methods*. 14, 403–406, 2017.). We understand that the performance of the traditional machine learning models may have improved if we tried many other feature descriptors. However, we believe that convolutional neural network has an advantage because feature extraction is not needed for training. We have added these results to the manuscript as described below:

“We compared these results with feature-based traditional machine learning methods (Support Vector Machine, Random forests, and logistic regression) to confirm the superiority of CNN. To analyse cellular images using classical machine learning models, we extracted features of images to create input datasets. We used Histograms of Oriented Gradients (HOG) which is one of the most commonly used feature descriptors and trained the machine learning models. The accuracy and F1 score of traditional machine learning models were lower than that of the CNN and we concluded that the CNN is the most suitable method for our study (Extended Data Fig. 3f).”

2. It is not clear how to calculate a quantitative senescence score based on the CNN predictions. For each input, the CNN outputs the probability of the input belonging to the i -th class as: $y_i = \exp(x_i) / \sum_j (\exp(x_j))$, as shown in Line 5 at Page 24. However, what does “ $\sum_i (y_i) / n$ ” mean in Line 7 at Page 24? What does n present? In addition, the senescence score is calculated as the average probabilities over all cells in a single input image or in the entire test set?

Thank you for your comments. As you pointed out, the methods are insufficient to fully understand how the senescence score was calculated. First, we cropped the input images from the entire test dataset (larger images) at single cell resolution by a defined method (Extended Data Fig. 2a, b). Each single image was input into the pre-trained CNN and then the probability of senescent cells was output. Then, we calculated the average probabilities over all cells in a single input image. Softmax: y_i is the probability for a single image (cell), and senescence score is the averaged probability for all images (cells) in each condition. In Line 5 on Page 24, “ i ” means i -th belonging classes in the softmax function. Our task was binary classification, therefore, the sigmoid function could be used in the last layer of the CNN, but we used the softmax function to build a system that enables multi-class classification in the future. In Line 7 on Page 24, “ i ” means the i -th input datasets. “ n ” is the total number of the input datasets obtained from larger images at the single cell resolution. Using same variable “ i ” in both line 5 and line 7 has obviously resulted in confusion, and we have therefore changed the variable description in line 7 from “ i ” to “ k ”. We have revised the sentence as shown below to convey our meaning more accurately:

“We calculated the senescence score using the probability of senescence, which represents the outputs

of trained classification networks to detect senescent cells, for each cell at the single cell resolution. The neural network was able to calculate the probability class of input data, when the softmax function was used in the output layer:

$$\text{softmax: } y_i = \frac{e^{x_i}}{\sum_{j=1}^N e^{x_j}}$$

The average probabilities of belonging to the senescence class in all input datasets were used for calculating the senescence score:

$$\text{senescence score} = \frac{\sum_{k=1}^m y_{1k}}{m}$$

(m : the number of input datasets, y_j : the probability of belonging to the senescence class in each input dataset) “

3. Can the proposed method be trained on a dataset but be used on another dataset that is from a different institution?

Thank you for your comments. We are also very interested in whether our CNN-based method can be applied to datasets obtained at other institutions. We therefore collaborated with a different institute, Kyoto University, and ask them to culture and take images of control and senescent HUVECs. First, we examined whether the CNN trained by the datasets at Keio University (our institution) could classify the Kyoto University datasets of control and senescent HUVECs. The CNN could moderately distinguish the cellular state, but the performance was lower than the validation results shown in Fig. 1f, g, and Extended Data Fig. 4a, b. Then, we combined the Keio and Kyoto datasets, and trained the CNN to further increase generalizability. The CNN was successfully trained on the both datasets with high performance, which suggests the trained CNN could work for the both Keio and Kyoto datasets (Extended Data Fig. 5a, b). We tested the performance of the CNN using the Kyoto datasets which were not used for training, and found the CNN trained on both institutes had a higher performance (Extended Data Fig. 5c). Importantly, the CNN also had a high performance on the Keio datasets, which suggests that the generalizability of the CNN trained on the datasets from both institutes is higher than that trained on a single institute dataset. Moreover, the senescent score created by Deep-SeSMo was not only well correlated with CPT or H₂O₂ concentration on the Kyoto datasets but also on the Keio datasets (Extended Data Fig. 7b, c). We conclude that our methods can be applied to datasets from other institutes, but the performance will increase if the CNN is trained on datasets from all institutes where we plan to validate data. We have added this information to the manuscript as shown below:

“Moreover, we examined whether the CNN can be applied to datasets obtained at another institution, Kyoto University. HUVECs were cultured and phase-contrast images were acquired at Kyoto University. The CNN was successfully trained on both the Keio (our institution) and Kyoto datasets with high performance (Extended Data Fig. 5a, b). We tested the performance of the CNN on Kyoto datasets which were not used for training, and found that the CNN trained on the datasets from both institutes have a higher performance (Extended Data Fig. 5c). Importantly, the CNN also has a high performance with the Keio datasets, which suggests that the CNN trained on datasets from both institutes has higher generalizability.”

“Senescence score which was generated by the CNN trained on the datasets acquired at two institutes, Keio and Kyoto (Extended Data Fig. 7b, c), or the CNN trained on another cell type, HDFs (Extended Data Fig. 7d, e) also showed high performance.”

4. An intensity thresholding method is used to crop cell patches for CNN training. Does the threshold value change for different images? Is it possible that this thresholding method generates noisy patches such that there are no cells inside these patches or the cells are not centered? If so, how does the proposed method deal with these cases? How will it perform in the presence of noisy data? How much noises it can handle?

Thank you for your comment. It is an important point to note how we deal with noisy data in an intensity thresholding method. We defined the threshold value for image binarization. The threshold value was not confirmed in every image, because we took each phase contrast image in almost same condition every time. However, we occasionally confirmed that the threshold value could correctly identify the cells, and cropped images correctly showed a single endothelial cell at the center of the image. As you noted, noises were also identified as black particles in the binarized images. However, noise particles are obviously smaller than the cell sizes. Therefore, we define the size under the cell size as noise, and the cropped images under the defined size are automatically eliminated for further analysis. Using this condition, we could exclude almost the small particles and minimize noisy data. We show excluded noisy data in Extended Data Fig. 2b. Moreover, Human Umbilical Vein Endothelial Cells (HUVECs) have a cobblestone-like morphology, making it easy to identify the center of the cell as a black particle in the binarized phase contrast images. Cropped images in which the cells were not centered were rare. Actually, a small amount of noisy data was included in the input dataset, but they were mostly fractions of damaged cells, which are often observed in the senescent condition, and identified as senescent cells. Thus, we think noisy data do not influence the performance of the CNN. We have revised the methods section as described below:

“The threshold value for cell size was determined, and we confirmed that the cells could be correctly identified. We also defined the noise particles as being smaller in size than the cells. Thus cropped images under the defined size were automatically eliminated for further analysis.”

5. Is it possible to directly link the discriminative regions identified by the Grad-CAM to the senescence score computation and thus improve the interpretability of Deep-SeSMo? How can these important regions be used to analyze the effects of anti-cellular senescence reagents or understand cellular senescence suppression (if applicable)?

Thank you for your attractive suggestion. In this study, we used the Grad-CAM to visualize the features on which the CNN was trained, and to confirm that the CNN would identify the correct features of senescent morphology. It would be highly significant if we could extract the unique features in important lesions presented by Grad-CAM and improve the performance of Deep-SeSMo. However, we were unsuccessful in using this method to improve performance of the CNN at this time. As you pointed out, Grad-CAM could be also used for analysis of anti-senescent candidate drugs. We believe that we should first elucidate why senescent cells differentially showed morphological changes around the cellular peripheral areas and focus on the heterogeneous intracellular areas in the future. These analyses would help us to understand the biology of senescence. We will try them in future studies to improve our deep learning-based methods.

6. How will incorrect predictions of the CNN affect the computation of the senescence score? It will be helpful to analyze/discuss the failures and the limits of the proposed method.

Thank you for your suggestion. We output images of true positive (prediction: senescence, answer: senescence), true negative (prediction: healthy, answer: healthy), false positive (prediction: senescence, answer: healthy), false negative (prediction: healthy, answer: senescence), presented in Extended Data Fig. 3g. The morphological appearance of false positive images was similar to true positive images, and false negative images were similar to true negative images to our eyes. This suggests that one limitation of our method might be the teaching data. We created teaching data as follows: label the cells cultured in healthy conditions as healthy cells, and label the cells cultured in senescence-inducing conditions as senescent cells. However, a very small proportion of senescent cells would exist under healthy conditions, and a very small proportion of healthy cells would exist under senescence-inducing conditions. Thus, incorrect prediction is absolutely unavoidable, even though we paid full attention to

the preparation of healthy or senescent cells. Although we would be able to detect the senescent cells present in healthy culture conditions by using immunostaining of senescent markers, it would add unnecessary complexity to our CNN-based method. However, in our current analyses, incorrect predictions of the CNN were very rare, therefore we believe that these incorrect predictions would have very little effect on the computation of the senescence score. We have revised the discussion as shown below:

“Although the CNN showed high performance, there were still mispredictions. When we output the false decision images (Extended Data Fig. 3g), the morphological appearance of false positive images was similar to that of true positive images, and false negative images were similar to true negative images. These suggest that a very small proportion of senescent cells exist in healthy conditions, and a very small proportion of healthy cells exist in senescence-inducing conditions, even though we paid full attention to the preparation of healthy or senescent cells. However, in our current analysis, incorrect predictions of the CNN were very rare, therefore we believe that any incorrect predictions would have very little effect on the computation of the senescence score.”

7. Which layer of the CNN uses dropout? What is the dropout rate?

We used the dropout technique following the layer of first and second Max pooling, and the first dense layer. The dropout rate was 0.5. This information is written in Extended Data Fig. 2e. We have also added this information in the methods section as shown below:

“To avoid overfitting, dropout techniques were used following the layer of first and second Max pooling, and the first dense layer. The dropout rate was 0.5.”

8. Are the input images normalized before feeding them to the CNN? If so, how?

We normalized input images as shown below:

Normalized input images = $((\text{original images} / 255) - 0.5) \times 2$

Max value of normalized images were “1”, and min value of normalized images were “-1”.

We have revised the methods section to include this information, as shown below:

“We normalized input images as follows:

Normalized input images = $((\text{original images} / 255) - 0.5) \times 2$

Max value of normalized images were “1”, and min value of normalized images were “-1”.”

Reviewer #2 (Remarks to the Author):

In their recent manuscript, Kusumoto et al. present a novel CNN (Deep-SeSMo) capable of automated image analysis to discriminate between senescent HUVECs and control endothelial cells under multiple senescence-inducing conditions (replicative stress, camptothecin, and peroxide). The authors validate Deep-SeSMo using several senescence-suppressing compounds before applying their technique to screen an 80-member kinase inhibitor library for potential novel senescence-suppressing molecules. These results are of great potential interest, particularly considering the growing interest in developing senolytic or senomorphic compounds for treatment of age-related diseases. However, I have several concerns that, should they be addressed, would greatly strengthen the manuscript.

Thank you very much for your careful reviewing and valuable comments. Based on your comments, we have performed additional experiments and analyses to improve the quality of our study, and have revised the manuscript accordingly.

1.) Clarification of the methods for drug treatments and screening. It is implied but not explicitly stated in the Methods or main text that senescent HUVECs were generated prior to treatment with either the candidate kinase inhibitor library and the hits from CNN screening. It is essential to clarify how senescent HUVECs were generated and when in the timeline of senescence induction the compounds were applied. It would also be important to test at least 4 compounds from the kinase inhibitor panel that did not show senomorphic activity based on Deep-SeSMo and show that these do not have an effect on p53-p21 axis activity. Furthermore, the authors do not examine effects on proliferative arrest or p16 expression, other major hallmarks of cellular senescence, either in their senescent cell cultures with or without senomorphic treatment used as the training datasets or the 4 kinase inhibitor hits identified by Deep-SeSMo.

Thank you for your comments. As you mentioned, we should clarify the timeline of the senescence induction and compound application in the manuscript. For the drug screening, we prepared the senescent HUVECs for three induction methods; H₂O₂, Camptothecin, and replication. For the 80 drugs tested, 10 μM of each drug was added at the same time as 0.15 mM H₂O₂ for four days, or 25 nM CPT for two days. For the replicative stress screens, moderately senescent HUVECs were incubated with drugs for five days. We have revised the methods section as shown below:

“For the drug screening or drug assessment in HUVECs, 0.15 mM H₂O₂ was added for four days or 25 nM CPT for two days, with simultaneous application of the test compounds. For the replicative

stress screens, moderately senescent HUVECs were incubated with compounds for five days. As a control for the drug treatments, the same concentration of dimethyl sulfoxide (DMSO) was used alongside senescence induction.”

According to your suggestion, we examined whether the compounds from the kinase inhibitor panel that did not show senomorphic activity based on Deep-SeSMo might still influence P53-P21 axis activity. We selected the non-effective drugs SC-514, TYRPHOSTIN51, Indirubin, and SU4312, which showed almost the same senescent score as the control by Deep-SeSMo analysis. We examined the P53-P21 axis activity of the drugs by western blotting and found that they have almost no effects on the activation of this pathway (Extended Data Fig. 9e). We have revised the manuscript as shown below:

“We also examined the effects of four drugs (SC-514, TYRPHOSTIN51, Indirubin, and SU4312, which were determined as the non-effective drugs by Deep-SeSMo analysis, with almost the same senescence score as the control) on the P53-P21 senescence axis. All four drugs showed almost no effects on the activation of the P53-P21 signalling pathway (Extended Data Fig. 9e).”

As you have mentioned, P16INK4a is an important hallmark of cellular senescence. We examined the expression of P16INK4A by western blotting, and confirmed that all four anti-senescent candidate drugs decreased P16INK4a expression (Fig. 4c). We have revised the manuscript to include this information, as shown below:

“Western blotting also demonstrated that the four compounds suppressed the P53-P21 axis activation and P16INK4a expression (Fig. 4c).”

2.) All presented validation and screening focuses on “senomorphic” compounds and in HUVECs specifically. Essentially, the authors focus on determining whether Deep-SeSMo is capable of detecting senescence-suppressing effects of known senomorphic compounds (such as metformin) and identifying novel compounds with similar effects. While this is indeed of significant value, a great deal of drug discovery focus within the senescence field is on senolytic (senescent cell killing) drug identification. It would be of considerable value if the authors could apply their CNN to determine whether known senolytics, such as ABT263 (navitoclax), are capable of depleting senescent HUVECs from a mixed culture of senescent and control cells.

Thank you for your comments. It is a very interesting suggestion to examine whether the Deep-SeSMo analysis could be applied to drug discovery for senolytics. We explored this possibility using ABT263 (Navitclax). We mixed young (1×10^5 cells/well) and old (2×10^5 cells/well) HUVECs, treated them with 0.25 μM , 0.5 μM , 1 μM or 2.5 μM of ABT263 for 72 hours, and analyzed the cells using Deep-SeSMo. Before ABT263 treatment, young and old HUVECs appeared to be successfully mixed in the phase-contrast microscopic images (Response Fig. 1a). After 72 hours, the images of HUVECs with no drugs (control) showed that both young and old HUVECs were still present. However, HUVECs exposed to ABT263 for 72 hours, especially with 1 μM ABT263, appeared to comprise mostly young cells in the phase contrast images. In our experimental conditions, the 2.5 μM concentration of ABT263 was too high to use as a senolytic drug, because young HUVECs were also damaged. We then analyzed the effects of ABT263 by Deep-SeSMo and confirmed that Deep-SeSMo could correctly assess the senolytic effect of ABT263 (Response Fig. 1a, b). It would be interesting to further examine and improve the validation of senolytic drugs in future using Deep-SeSMo. Again, thank you for your wonderful and interesting suggestion.

Response Figure 1

Response Figure 1: Deep-SeSMo for Senolytics.

a, Representative images of HUVECs, which were a mixture of young and old cells, treated with several concentrations of ABT263 for 72 hours. **b**, Senescence score calculated by Deep-SeSMo showed the senolytic effect of ABT263.

Additionally, all observations based on Deep-SeSMo are limited to HUVECs in monolayer culture. While this is certainly important proof of principle that this image analysis approach is viable for drug screening for senomorphics in senescent cell culture, it would be encouraging to know that a CNN

developed on similar principles would be able to, at least, discriminate between senescent and control cells of a non-endothelial origin, such as human diploid fibroblasts or mouse embryonic fibroblasts. I suggest these cell types because they also reliably respond to peroxide, DNA-damaging agents, and extended passaging with the senescence response.

Thank you for your comments. As you mentioned, it is important to examine whether our system can be applied to other cell types. We therefore induced cellular senescence in the human diploid fibroblast (HDF) cell line, TIG114, and trained the CNN as described for HUVECs. We added 0.02 mM H₂O₂ to the HDFs for four days, and 200 nM CPT for three days to induce cellular senescence. We repeated two independent inductions with both conditions and acquired phase contrast images (Extended Data Fig. 5d). The CNN was successfully trained with a high performance: the F1 score was 0.96, the accuracy was 0.97, and the AUC was near 1.0 (Extended Data Fig. 5e). Then, we tested the CNN performance using newly acquired datasets, and confirmed that the CNN has produced high performance in the test datasets (Extended Data Fig. 5f). Moreover, we calculated the senescence score by using the HDF-trained CNN and demonstrated that the senescence score was well correlated with H₂O₂ or CPT concentration (Extended Data Fig. 7d). Thus, we concluded that our CNN-based method can be applied to other cell types. Interestingly, CNN trained on HUVEC datasets was also able to classify the healthy and senescent HDFs (Extended Data Fig. 5g), and showed high performance in producing a senescent score using the HDF datasets (Extended Data Fig. 7e), indicating that the CNN was able to recognize the common morphology of cellular senescence across the cell types. We have therefore revised our manuscript as shown below:

“We also examined whether CNN could classify senescence in other cell types. We used human diploid fibroblasts (HDFs), induced cellular senescence by H₂O₂ or CPT, cropped input datasets at single cell resolution levels, and trained the CNN to classify them (Extended Data Fig. 5d). The CNN was successfully trained (Extended Data Fig. 5e), and had a high performance in the test datasets (Extended Data Fig. 5f). Interestingly, the CNN trained on HUVEC-datasets was also able to classify healthy and senescent HDFs (Extended Data Fig. 5g).”

“the CNN trained on another cell type, HDFs (Extended Data Fig. 7d, e), also showed high performance.”

In summary, the authors have produced a potentially very useful approach to automated identification of senescent cells at the single cell level using morphology alone. However, this tool was only validated using HUVECs; several other classic markers of cellular senescence (such as p16 and

proliferative status) were not assessed; and, non-senomorphic hits from the kinase inhibitor screen were not checked to show that they do not alter molecular parameters of senescence in HUVECs. These are the most important limitations of the current study and addressing them would greatly increase confidence that Deep-SeSMo, or a related CNN, would have broad utility in identifying senomorphic compounds irrespective of cell type. Additionally, testing to see whether Deep SeSMo can also detect senescent cell killing (senolytic) compounds would enhance confidence that Deep SeSMo would be useful within the current senescence/drug discovery initiatives.

Thank you again for your comments and suggestions. We have conducted additional experiments and analysed them according to your suggestions, and have revised several aspects of our manuscript. We believe these revisions have improved the quality of our manuscript.

Reviewer #3 (Remarks to the Author):

In this manuscript, Kusumoto et al., describe anti-senescent drug screening by deep learning- based morphology senescence scoring. Specifically, authors developed a new convolutional neural networks (CNN)- based technique of senescence scoring. The technique termed as “Deep Learning-Based Senescence Scoring System” (Deep-SeSMo) appears to have a great potential of advancing drug - discovery, particularly in the area of ageing and cancer. The authors established this innovative method based on senescent phenotype observed in endothelial cells. Furthermore, authors successfully identified four novel anti-senescent drugs using this method. The utility of this innovative approach is validated further analysis of the compounds, which appear to inhibit a well-known inflammatory response pathway, otherwise known as “Senescence-Associated Secretory Phenotype” (SASP), a hallmark of the senescent cells. It is a well-done study, and a well-written manuscript. It should be of very interest to ageing and cancer researchers as well as pharma companies. Authors have used multiple types of stress agents that are known to induce premature senescence in human umbilical vein endothelial cells (HUVECs). The senescent phenotype scored by CNN based scoring methods has been verified by known markers of senescence such as SA-b-gal. The minor concern is about the wider applicability of the method to cell types other than HUVECs. The morphological features of senescence may vary in different cell types. Authors could further validate Deep-SeSMo using a different cell type such as a strain of human diploid fibroblast (HDF). The other specific comments are as following-

Thank you very much for your careful reviewing and valuable comments. Based on your comment, we have performed additional experiments and analyses to improve the quality of our study, and have revised the manuscript accordingly.

As you have mentioned, it is very important to validate that our method can be applied to other cell types. We therefore induced cellular senescence in a human diploid fibroblast (HDF) cell line, TIG114, and trained the CNN as described for HUVECs. We added 0.02 mM H₂O₂ to the HDFs for four days, and 200 nM CPT for three days to induce cellular senescence. We repeated two independent inductions for both conditions and acquired phase contrast images (Extended Data Fig. 5d). The CNN was successfully trained with a high performance: the F1 score was 0.96, the accuracy was 0.97, and the AUC was near 1.0 (Extended Data Fig. 5e). Then, we tested the CNN performance using newly acquired datasets, and confirmed that the CNN has also high performance in the test datasets (Extended Data Fig. 5f). Moreover, we calculated the senescence score by using the HDF-trained CNN and demonstrated that the senescence score was well correlated with H₂O₂ or CPT concentration (Extended Data Fig. 7d). Thus, we concluded that our CNN-based method can be applied to other cell

types. Interestingly, the CNN trained on HUVEC datasets was also able to classify healthy and senescent HDFs (Extended Data Fig. 5g), and showed high performance in producing a senescence score using the HDF datasets (Extended Data Fig. 7e), indicating that the CNN was able to recognize the common morphology of cellular senescence across the cell types. We have therefore revised our manuscript as shown below:

“We also examined whether the CNN could also classify senescence in other cell types. We used human diploid fibroblasts (HDFs), induced cellular senescence by H₂O₂ or CPT, cropped input datasets at single cell resolution levels, and trained the CNN to classify them (Extended Data Fig. 5d). The CNN was successfully trained (Extended Data Fig. 5e), and had a high performance in the test datasets (Extended Data Fig. 5f). Interestingly, the CNN trained on HUVEC-datasets was also able to classify healthy and senescent HDFs (Extended Data Fig. 5g).”

“the CNN trained on another cell type, HDFs (Extended Data Fig. 7d, e). also showed high performance.”

1. Results- Page 5 line 6, senescent cells are resistant to apoptosis, it is incorrect to say p21-p53 pathway induces apoptosis in senescent cells; hence remove apoptosis from sentence “p21-P53 pathway---- cell cycle arrest and apoptosis”.

Thank you for your comments.

We have revised the manuscript as shown below:

“In senescent cells, the P21-P53 pathway is activated to induce cell cycle arrest.”

2. Figure 2J and K- the graph indicates three replicates each for H₂O₂, CPT and Rep, however only one replicate is shown for Rep in heat map, and only one replicate indicated for Rep in the figure legend.

Thank you for your comments. As you have pointed out, we validated the senescence score in the replicative stress-induced cellular senescence experiment for one replicate. However, those datasets were evaluated by three CNNs, trained on H₂O₂, CPT, or both H₂O₂ and CPT, thus the heatmap shown in Fig. 2j showed nine values for the H₂O₂ and CPT conditions, and three values for the replicative stress condition. The graph in Fig. 2k showed three bars because they represent the average value

evaluated by three trained CNNs. We have revised the Figure legend as shown below:

“A heatmap shows Pearson correlations for each set of test data. Three independent experiments of H₂O₂- and CPT-induced senescence and one experiment for replication-induced senescence, evaluated by three trained CNNs, were conducted.”

3. Fig. 3 and 4- Please indicate how was senescence induced in control cells?

Thank you for your comments. We induced cellular senescence in HUVECs by H₂O₂ in Fig. 3a-e and Fig. 4a-c, and by three methods in Fig. 3f-i. For RNA sequence analysis in Fig. 4d-i, we used moderately senescent HUVECs. For the 80 drugs tested, 10 μM of each drug was added at the same time as 0.15 mM H₂O₂ for four days, or 25 nM CPT for two days. For the replicative stress screens, moderately senescent HUVECs were incubated with drugs for five days. For the control condition, we added Dimethyl sulfoxide (DMSO) to the medium at a 1/1000 dilution, because the 80 drugs were dissolved in DMSO and added at a 1/1000 dilution. We have revised the methods section as shown below:

“For the drug screening or drug assessment in HUVECs, 0.15 mM H₂O₂ was added for four days or 25 nM CPT for two days, with simultaneous application of the test compounds. For the replicative stress screens, moderately senescent HUVECs were incubated with compounds for five days. For RNA sequence analysis, we used moderately senescent HUVECs. As a control for the drug treatments, the same concentration of dimethyl sulfoxide (DMSO) was used alongside senescence induction.”

4. Figure 3C, Fig. 4C- it would be good to include p16INK4a in these Westerns.

Thank you for your comments. P16INK4a is an important hallmark of cellular senescence, and we agree that we should examine whether drugs can reduce P16INK4a expression by western blotting. We have added experiments to show that metformin and NMN decreased P16INK4a expression (Fig. 3c), and that all four anti-senescent candidate drugs decreased the expression of P16INK4a (Fig. 4c). We have revised the manuscript as shown below:

“NMN and metformin decreased the SA-β-gal positive cell ratio, P21-P53 activation, and P16INK4a expression.”

“Western blotting also demonstrated that the four compounds suppressed the P53-P21 axis activation and P16INK4a expression.”

Reviewers' Comments:

Reviewer #1:

Remarks to the Author:

The revised manuscript has addressed the reviewer's concerns, so the reviewer recommends acceptance of the paper.

Reviewer #2:

Remarks to the Author:

In their revised manuscript, Kusumoto et al. have performed new experiments in order to address my comments from the initial round of review. Essentially, my concerns focused on the areas of potential improvement listed below.

- 1.) Clarification of the experimental protocols for senescence induction and drug treatment. The authors have addressed this fully with textual changes.
- 2.) Demonstration that "non-hit" kinase inhibitors do not alter p53-p21 axis activation and demonstration that candidate senomorphics identified by their CNN do reduce markers of cellular senescence in vitro. The authors test 4 (hopefully random) non-hits and demonstrate a lack of alteration in p53-p21 engagement by Western blotting. Furthermore, for their 4 candidate hits (Figure 4C) that show a reduction in p53 and p21, they extend the observation by probing for p16Ink4a and show a reduction by Western. Collectively, these findings satisfy my concern.
- 3.) Determine whether their CNN successfully identifies known senolytics (such as Navitoclax) using senescent HUVECs. The authors test navitoclax with their CNN and, indeed, show a dose-dependent reduction in "senescence score" with ABT263. This satisfies my concern, and the authors/editor should consider the inclusion of Response Figure 1 as a control, at least in the Supplement.
- 4.) Determine whether their CNN can be trained to detect senescence-modifying compounds in other cell types. The authors apply their existing (HUVEC trained) CNN to senescent TIG114 HDF, as well as train a new CNN on this particular cell type. In both cases, the authors demonstrate that their CNNs are capable of discriminating between senescent and non-senescent HDFs. While it would of course be nice to see the same type of drug discovery approach applied with a novel cell type, this is outside the scope and was not specifically recommended in the initial round of review. The authors' efforts here address the concern.

In conclusion, the authors have taken my comments seriously and successfully addressed my critiques. The manuscript is stronger as a result.

Reviewer #3:

Remarks to the Author:

The revised manuscript is scientifically and technically sound. Authors answered all the critiques and performed additional experiments as suggested. The novel technique described as Deep-SeSMo should facilitate new discoveries related to senescence and aging.

Point to point response

Reviewer #1 (Remarks to the Author):

The revised manuscript has addressed the reviewer's concerns, so the reviewer recommends acceptance of the paper.

I'm really appreciate your comments and suggestions, because the quality of this study and manuscript has improved.

Reviewer #2 (Remarks to the Author):

In their revised manuscript, Kusumoto et al. have performed new experiments in order to address my comments from the initial round of review. Essentially, my concerns focused on the areas of potential improvement listed below.

I'm really appreciate your comments and suggestions, because the quality of this study and manuscript has improved.

1.) Clarification of the experimental protocols for senescence induction and drug treatment. The authors have addressed this fully with textual changes.

Thank you for your comment.

2.) Demonstration that “non-hit” kinase inhibitors do not alter p53-p21 axis activation and demonstration that candidate senomorphics identified by their CNN do reduce markers of cellular senescence in vitro. The authors test 4 (hopefully random) non-hits and demonstrate a lack of alteration in p53-p21 engagement by Western blotting. Furthermore, for their 4 candidate hits (Figure 4C) that show a reduction in p53 and p21, they extend the observation by probing for p16Ink4a and show a reduction by Western. Collectively, these findings satisfy my concern.

Thank you for your comment. We chose four non-hit drugs at random based on senescence score.

3.) Determine whether their CNN successfully identifies known senolytics (such as Navitoclax) using senescent HUVECs. The authors test navitoclax with their CNN and, indeed, show a dose-dependent reduction in “senescence score” with ABT263. This satisfies my concern, and the

authors/editor should consider the inclusion of Response Figure 1 as a control, at least in the Supplement.

Thank you for your comment. We showed this result in supplementary figure 7f and 7g. We revised the manuscript as shown below:

“Senolytics are focused as potential therapeutic drugs for age-related diseases, to induce apoptosis specifically in senescent cells¹⁷. We examined whether Deep-SeSMo could correctly assess the senolytic effect of ABT263. We mixed the young and old HUVECs, treated them with ABT263, and analysed the cells using Deep-SeSMo. Deep-SeSMo could also correctly assessed the senolytic effect of ABT263 (Supplementary Fig. 7f, g).”

“Senolytics treatment

We mixed young (100,000 cells / well) and old HUVECs (200,000 cells/ well) one day before drug treatment. 0.25 mM, 0.5 mM, 1 mM and 2.5 mM of ABT263 (Adoap) were added to the HUVECs for three days. Five phase-contrast images were acquired in each condition, including before drug treatment, and senescence score was calculated using Deep-SeSMo. “

4.) Determine whether their CNN can be trained to detect senescence-modifying compounds in other cell types. The authors apply their existing (HUVEC trained) CNN to senescent TIG114 HDF, as well as train a new CNN on this particular cell type. In both cases, the authors demonstrate that their CNNs are capable of discriminating between senescent and non-senescent HDFs. While it would of course be nice to see the same type of drug discovery approach applied with a novel cell type, this is outside the scope and was not specifically recommended in the initial round of review. The authors’ efforts here address the concern.

Thank you for your comment.

In conclusion, the authors have taken my comments seriously and successfully addressed my critiques. The manuscript is stronger as a result.

Thank you again for your comments.

Reviewer #3 (Remarks to the Author):

The revised manuscript is scientifically and technically sound. Authors answered all the critiques and performed additional experiments as suggested. The novel technique described as Deep-SeSMo should facilitate new discoveries related to senescence and aging.

I'm really appreciate your comments and suggestions, because the quality of this study and manuscript has improved.